# Reactive oxygen species affect the potential for mineralization processes in permeable intertidal flats

Marit R. van Erk [1,3,5] ✉, Olivia M. Bourceau [1,5] ✉, Chyrene Moncada [1], Subhajit Basu[1,4], Colleen M. Hansel [2] & Dirk de Beer [1]

Intertidal permeable sediments are crucial sites of organic matter remineralization. These sediments likely have a large capacity to produce reactive oxygen species (ROS) because of shifting oxic-anoxic interfaces and intense iron-sulfur cycling. Here, we show that high concentrations of the ROS hydrogen peroxide are present in intertidal sediments using microsensors, and chemiluminescent analysis on extracted porewater. We furthermore investigate the effect of ROS on potential rates of microbial degradation processes in intertidal surface sediments after transient oxygenation, using slurries that transitioned from oxic to anoxic conditions. Enzymatic removal of ROS strongly increases rates of aerobic respiration, sulfate reduction and hydrogen accumulation. We conclude that ROS are formed in sediments, and subsequently moderate microbial mineralization process rates. Although sulfate reduction is completely inhibited in the oxic period, it resumes immediately upon anoxia. This study demonstrates the strong effects of ROS and transient oxygenation on the biogeochemistry of intertidal sediments.

Reactive oxygen species (ROS) are short-lived oxygen-containing intermediates with lifetimes of seconds to hours, including superoxide, hydrogen peroxide and hydroxyl radicals. They are formed by a variety of photochemical, abiotic, and biotic processes[1]. Biotic formation occurs both intracellularly and extracellularly as a byproduct of metabolic and other physiological mechanisms[2]. In addition to photochemical pathways, a number of light-independent abiotic processes can lead to ROS formation, including oxidation of sulfide and ferrous iron ($Fe^{2+}$)[3,4], as well as anaerobic reactions with pyrite[5]. Intracellular ROS can damage cell components such as DNA, proteins, and lipids via a range of oxidative processes[6], and thus be detrimental to microorganisms at elevated levels. However, both intracellular and extracellular ROS also have beneficial roles, including pathogen resistance[7], nutrient acquisition[8], microbial growth[9], and as signaling molecules[10]. As such, ROS levels are strictly controlled by degrading enzymes[2], such as superoxide dismutase, which converts superoxide to hydrogen peroxide, and catalase, which converts hydrogen peroxide to oxygen and water. Electron donor-driven mechanisms also actively degrade ROS, such as through reactions with metals and organic material[11].

Despite the large potential of ROS to influence microbial processes, the distribution of ROS, including hydrogen peroxide, in marine sediment is understudied. To date, only a few studies have investigated hydrogen peroxide concentrations in sediments[12], and most of these focused on the potential of sediments to generate hydrogen peroxide upon oxygenation or sulfide exposure[3]. Studies of anoxic soils and aquifer sediments have shown great potential for the generation of ROS upon reoxygenation and have also shown that ROS directly impacts $CO_2$ evolution[13–17]. As hydrogen peroxide has also been demonstrated to have both stimulatory and inhibitory effects on microorganisms[6,7,10], it may greatly affect carbon cycling in marine sediments.

[1]Max Planck Institute for Marine Microbiology, Bremen, Germany. [2]Department of Marine Chemistry and Geochemistry, Woods Hole Oceanographic Institution, Woods Hole, MA, USA. [3]Present address: Department of Earth Sciences, Utrecht University, Utrecht, The Netherlands. [4]Present address: School of Health Sciences and Technology (SoHST), University of Petroleum and Energy Studies (UPES), Dehradun, Uttarakhand 248007, India. [5]These authors contributed equally: Marit R. van Erk, Olivia M. Bourceau. ✉e-mail: merk@mpi-bremen.de; obourcea@mpi-bremen.de

Particularly during disturbance events and at oxic–anoxic interfaces, which happen frequently in intertidal permeable sediments, elevated ROS levels are expected[12,16–20]. The depth to which oxygen penetrates into intertidal permeable sediment varies according to tides, currents, storms, and bioturbation[21]. The oxic zone can shift between several mm to several cm deep multiple times a day[22]. Nevertheless, anaerobes in the upper sediment maintain high rates of sulfate reduction, dissimilatory nitrate reduction, fermentation, and other anaerobic processes[23–25]. The high rates of carbon and nitrogen remineralization make these sediments biocatalytic filters[21,26], essential for the functioning of shallow water ecosystems. Consequently, ROS may play an unappreciated role in the biogeochemistry of dynamic coastal sediments.

This work supports our hypothesis that high levels of ROS develop in intertidal permeable sediments and that ROS have the potential to control biomineralization rates. Both a newly developed $Fe^{2+}$-resistant hydrogen peroxide microsensor and a chemiluminescence method detect significant concentrations of hydrogen peroxide in intact sediment cores and extracted porewater from the intertidal sandflat Janssand in the German Wadden Sea. Potential biogeochemical process rates in slurries of these sediments amended with ROS-removing enzymes confirm that ROS can affect microbial respiration. Potential rates of oxygen consumption, sulfate reduction, and $H_2$ and dissolved $Fe^{2+}$ accumulation are all increased by the removal of ROS. This impact is despite the varied recovery time after oxygenation, with sulfate reduction resuming immediately but $Fe^{2+}$ accumulation taking more than 12 h. The distribution and impact of ROS in these extremely dynamic environments deserve further attention, as ROS may well be important in coastal carbon cycling.

## Results and discussion

### Hydrogen peroxide in intertidal sediments

We determined porewater hydrogen peroxide concentrations in sandy surface sediments from the intertidal sandflat Janssand (53°44'25.51"N, 7°41'28.63"E)[27–29], a sandflat in the German Wadden Sea. Hydrogen peroxide concentrations were determined using two independent methods: microsensing in a sediment core, and a chemiluminescence technique applied to porewater that was extracted from either a parallel sediment core, or directly from the flat. Microprofiles of hydrogen peroxide, which were measured with a newly developed microsensor, show the presence of hydrogen peroxide (Fig. 1a, b and Supplementary Fig. 1a–c). The microsensors that were used contained ferrozine in their electrolyte, and were therefore not sensitive to $Fe^{2+}$ (Supplementary Fig. 2), which is abundant in these sediments (Supplementary Fig. 3). Steady-state hydrogen peroxide levels were elevated in comparison to many environmental systems[13,14,17,30], reaching concentrations >50 μM. Maximum hydrogen peroxide production, determined from the microprofiles, was $1 \times 10^{-4}$ mol m$^{-3}$ s$^{-1}$ (Supplementary Fig. 4),

which is much higher than in tidal pools, soil waters, aquifers, and brackish and freshwater ponds[13,14,18,31].

The presence of hydrogen peroxide was confirmed using a chemiluminescence technique. Hydrogen peroxide was measured in porewater extracted from a sediment core not used for microsensor measurements, and in porewater extracted on the sandflat. Porewater extracted on the sandflat was directly fixed with ferrozine in the extraction syringe to prevent reaction with iron before analysis. Absolute concentrations determined by chemiluminescence (Supplementary Fig. 5a, b) differ from those in microprofiles, which may be explained by sampling artefacts and loss during storage between sampling and analysis. As the samples measured using chemiluminescence were not fixed with acid, loss of hydrogen peroxide may have occurred between sampling and analysis a few hours later. Nevertheless, both methods detected large quantities of hydrogen peroxide.

At steady state, the standing stock of hydrogen peroxide represents a balance between production and consumption processes. Injection of oxygenated seawater at 3 and 4 cm depth led to a transient peak of hydrogen peroxide (Fig. 2a, b and Supplementary Fig. 6a). Processes responsible for hydrogen peroxide in deeper, anoxic sediments may be related to cycling of dissolved $Fe^{2+}$, iron oxides, and pyrite[3–5]. The sediments were iron-rich, with concentrations of $9.7 \pm 1.8$ and $2.3 \pm 0.6$ μmol g$^{-1}$ sed in surface sediments (0–2 cm depth) in May and July 2020, and $4.0 \pm 0.3$ μmol g$^{-1}$ sed in deep sediment (10–14 cm depth) in July 2020 (Supplementary Table 1), and $Fe^{2+}$ was present in the porewater (Supplementary Fig. 3). Despite persistent hydrogen peroxide levels at some depths, the sediment had a high capacity to rapidly degrade hydrogen peroxide. Additions of hydrogen peroxide resulted in rapid consumption at 3 and 4 cm depth (Fig. 2c and Supplementary Fig. 6b, c), as the transient peaks of hydrogen peroxide lasted for only less than a minute. Injections of hydrogen peroxide near an oxygen sensor at the same depths resulted in large transient oxygen peaks (Fig. 2c and Supplementary Fig. 6d). The underlying mechanisms responsible for the hydrogen peroxide-induced burst in oxygen are currently unresolved, but may include rapid catalytic activity by catalase within and outside the cell. We conclude that the ROS concentration in these sediments is delicately balanced between production and consumption.

### Impact of reactive oxygen species on respiration

To test if the ROS in porewater can affect respiration, we used constantly mixed sediment slurries with several different amendments of ROS-removing enzymes catalase and superoxide dismutase, both together and alone. Some sediments were amended with an inert (i.e., non-ROS degrading protein, bovine serum albumin (BSA)). BSA served as a protein control, to rule out any non-catalytic effects of protein in the catalase and superoxide dismutase amendments. The effect of the amendments on aerobic respiration and sulfate reduction, the most

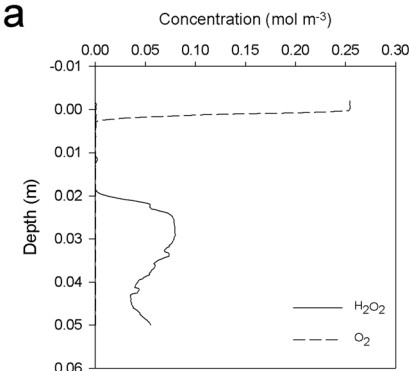
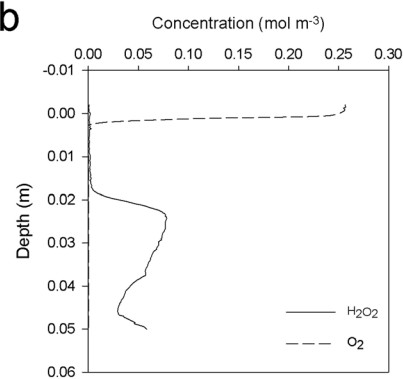

**Fig. 1 | Steady-state microprofiles of hydrogen peroxide and oxygen. a** Microprofiles ($H_2O_2$; solid line, $O_2$; dashed line) measured at 17:00. **b** Same measurements repeated at 20:30.

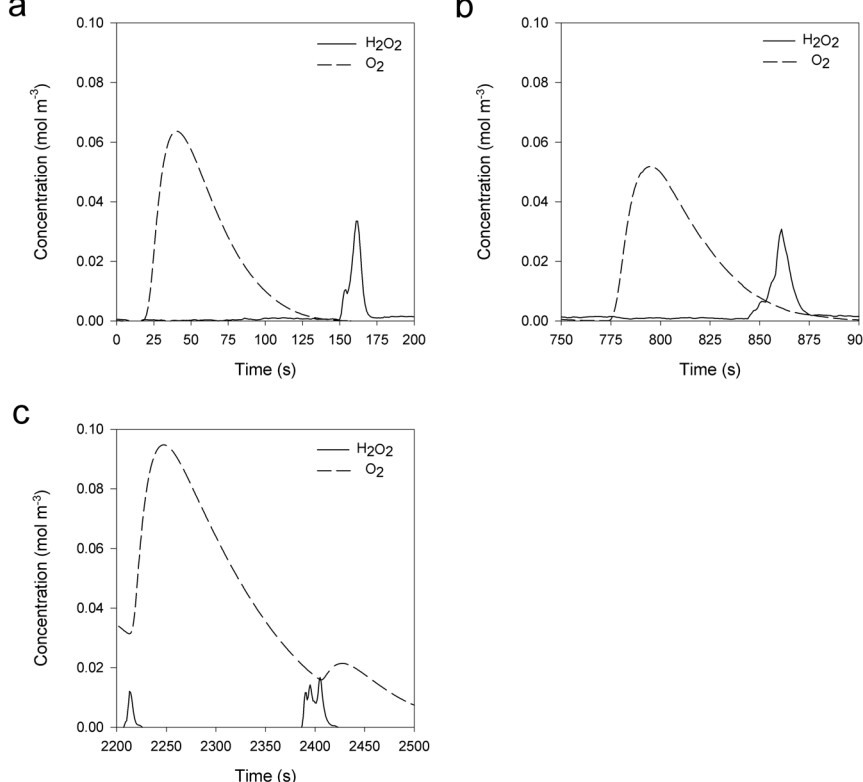

**Fig. 2 | Dynamics of hydrogen peroxide (H$_2$O$_2$; solid line) and oxygen (O$_2$; dashed line) in a sediment core.** Hydrogen peroxide and oxygen microsensors were at a constant position (4 cm depth). **a**, **b** Hydrogen peroxide and oxygen concentrations after injection of oxygenated seawater, **c** hydrogen peroxide and oxygen concentrations after injection of hydrogen peroxide.

relevant processes for coastal carbon turnover[32], and on hydrogen and Fe$^{2+}$ accumulation was assessed by using oxygen microsensing, a $^{35}$S-SO$_4$$^{2-}$ radiotracer technique, gas chromatography, and spectrophotometry, respectively. In all cases, the slurries were initially oxic and became anoxic within the first hours of incubation.

While slurries do not completely mimic the natural heterogeneity and complexity of sediment environments, they have the advantage of more uniform environmental conditions than intact sediment cores, thus allowing for measurements along a time series[33]. Since we were interested in respiration rates over an oxic–anoxic transition, and in the effects of transient ROS exposure, it was important to be able to precisely determine the oxic–anoxic shift. All rates presented here are potential conversion rates rather than environmental rates, although measured rates are within previously reported ranges for intertidal flat sediments[25,34].

Removal of hydrogen peroxide and superoxide via additions of catalase and superoxide dismutase, respectively, substantially increased the rates of oxygen consumption, sulfate reduction, and Fe$^{2+}$ and hydrogen accumulation (Fig. 3). This effect was not simply due to the addition of protein as a carbon or nitrogen substrate, as incubations with a comparable amount of BSA, did not stimulate these biogeochemical process rates. Superoxide dismutase and catalase both enzymatically produce oxygen as follows (reactions 1 and 2, respectively):

$$2O_2^{\bullet-} + 2H^+ \rightarrow H_2O_2 + O_2 \tag{1}$$

$$2H_2O_2 \rightarrow 2H_2O + O_2 \tag{2}$$

Despite this enzymatic production/recycling of oxygen, oxygen consumption was approximately four times faster in incubations containing both catalase and superoxide dismutase, showing ROS affects aerobic respiration in these sediments (Fig. 3a).

Sulfate reduction, for which rates were calculated for the anoxic period of the incubations, was faster in the presence of a combination of catalase and superoxide dismutase (Fig. 3b). The effect of ROS removal on carbon turnover rates was calculated using rates for untreated slurries and slurries amended with the combination of catalase and superoxide dismutase, using aerobic respiration (stoichiometry oxygen:carbon of 138:106) and sulfate reduction rates (stoichiometry sulfate:carbon of 1:2). Removal of ROS by catalase and superoxide dismutase led to a 4-times increase in biotic carbon turnover by aerobic respiration and sulfate reduction (Supplementary Table 2). We found no evidence that superoxide dismutase could alone affect respiration processes. This does not necessarily mean that superoxide does not impact the rate of sulfate reduction. In the absence of catalase, a positive effect of removing superoxide on the resident sulfate-reducing community may be masked by the negative effect of increasing hydrogen peroxide (reaction 1). Furthermore, superoxide dismutase is inhibited by hydrogen peroxide, which may have also masked any effect of superoxide[35]. Nevertheless, our data indicates that the addition of catalase, rather than superoxide dismutase, seems to be the driver for the increase of process rates after ROS removal.

Hydrogen and Fe$^{2+}$ accumulation rates were much higher in the presence of catalase alone and with combined catalase and superoxide dismutase (Fig. 3c, d). Fe$^{2+}$ accumulation could not be explained by the release of Fe$^{2+}$ from catalase, as the release of the 4 Fe atoms from the active site of each catalase molecule would only lead to a concentration of 0.5 μM. Net accumulation of Fe$^{2+}$, as shown in Fig. 3d, only

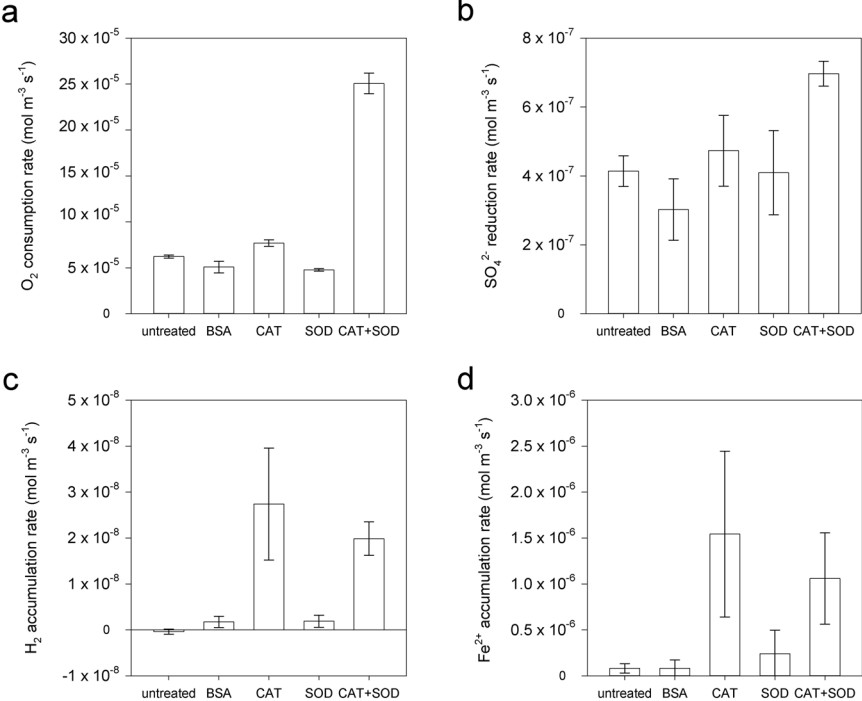

**Fig. 3 | The influence of reactive oxygen species (ROS) on respiration rates.** Respiration rates in untreated slurries and slurries treated with bovine serum albumin (BSA), and with ROS-removing enzymes catalase (CAT), superoxide dismutase (SOD), and a combination of CAT and SOD (CAT + SOD), from sediments collected on June 15, 2020. **a** Oxygen ($O_2$) consumption, **b** sulfate ($SO_4^{2-}$) reduction, **c** hydrogen ($H_2$) accumulation, **d** ferrous iron ($Fe^{2+}$) accumulation. Sulfate reduction, hydrogen accumulation, and $Fe^{2+}$ accumulation rates were calculated for the anoxic period of the incubation. Error bars represent the standard error of the slope.

occurred under anoxic conditions. Any oxygen produced by the activity of catalase and superoxide dismutase under these anoxic conditions will directly be used for the reoxidation of reduced compounds such as $Fe^{2+}$ or sulfide. Production of oxygen by enzyme activity is thus likely slower than the production of reduced compounds. In situ, hydrogen concentrations were low ($0.03 \pm 0.01$ nmol $cm^{-3}$ sed ($9.8 \pm 3.9$ nmol $L^{-1}$ porewater) in July 2020 and $0.04 \pm 0.01$ nmol $cm^{-3}$ sed ($13.1 \pm 2.1$ nmol $L^{-1}$ porewater) in March 2021), without trends with depth (Supplementary Table 3). Hydrogen in sediments can derive from fermentation, via $H_2S$ reacting with iron sulfide (Wächterhäuser reaction)[36,37], or $N_2$ fixation. Given the generally low-to-undetectable levels of dissolved sulfide in these sediments (Supplementary Table 4), we attribute most hydrogen production in our incubations to fermentation. Our results reconcile with previous observations that catalase additions significantly increase the rate of fermentation in cultures[38,39], but reveals a previously unknown constraint of ROS on fermentation in marine sediments.

Marine sediments are characterized by a tight balance between hydrogen production by fermentation and consumption by sulfate reduction[40]. To test if hydrogenotrophic sulfate reduction was active in these sediments, and if the presence of ROS could affect the tight balance, we amended slurries with sodium molybdate, a selective inhibitor of sulfate reduction. The ensuing hydrogen evolution (Supplementary Fig. 7) confirmed that hydrogenotrophic sulfate reduction occurred in these sediments. However, hydrogen fuels only a minor proportion of the sulfate reduction, as the highest hydrogen accumulation rate (with molybdate addition) was 20 times lower than the sulfate reduction rates (Supplementary Fig. 8). The addition of catalase alone and combined catalase and superoxide dismutase nevertheless disrupted the tight balance between hydrogen production by fermentation and consumption by sulfate reduction common in marine sediments[40] such that hydrogen accumulated.

## Anaerobic respiration after transient oxygenation
Oxygen and ROS sensitivity are expected to be closely linked, as much of the damage oxygen does to anaerobes is mediated by ROS[41]. Anaerobes are often regarded as extremely sensitive to oxygen, with their respiration taking several hours to recover from oxygenation. In their active state, reduced enzymes with iron cofactors that are involved in sulfate reduction become irreversibly damaged by oxygen, and release intracellular ROS, which can lead to cell death[6,42]. In an environment that frequently switches between oxic and anoxic conditions, such a degree of sensitivity would strongly suppress overall anaerobic respiration. Indeed, oxygen-tolerant anaerobic respiration has increasingly been measured in diverse environments[43–47]. We therefore investigated if there was a lag in the onset of anaerobic respiration after transient oxygenation in constantly mixed sediment slurries that were allowed to become anoxic.

An environment that fluctuates multiple times per day between oxic and anoxic conditions, such as surface intertidal sediments, exerts a strong selective pressure for sulfate reducers capable of coping with oxygen. However, the strong selective pressure for sulfate reducers to be able to withstand oxygen appears not to have selected for sulfate reducers capable of respiring in the presence of oxygen, as has been found in cyanobacterial mats[48]. Instead, though sulfate reducers were unable to perform sulfate reduction under oxic conditions, they performed sulfate reduction directly upon anoxia (Fig. 4a and Supplementary Fig. 9). While undetectable in the oxic period of incubations, sulfate reduction resumed instantly after oxygen depletion in both surface sediment normally subject to daily reoxygenation (0–2 cm depth) and deeper permanently anoxic sediment (10–14 cm depth) (Fig. 4a and Supplementary Fig. 9). Rapid sulfide oxidation to sulfate could not explain the absence of sulfate reduction during the oxic period, as it was also not detected by the silver wire method (Fig. 4b). That is the method of choice for detecting aerobic sulfate reduction since sulfide binds instantly and irreversibly to silver[49]. Incomplete

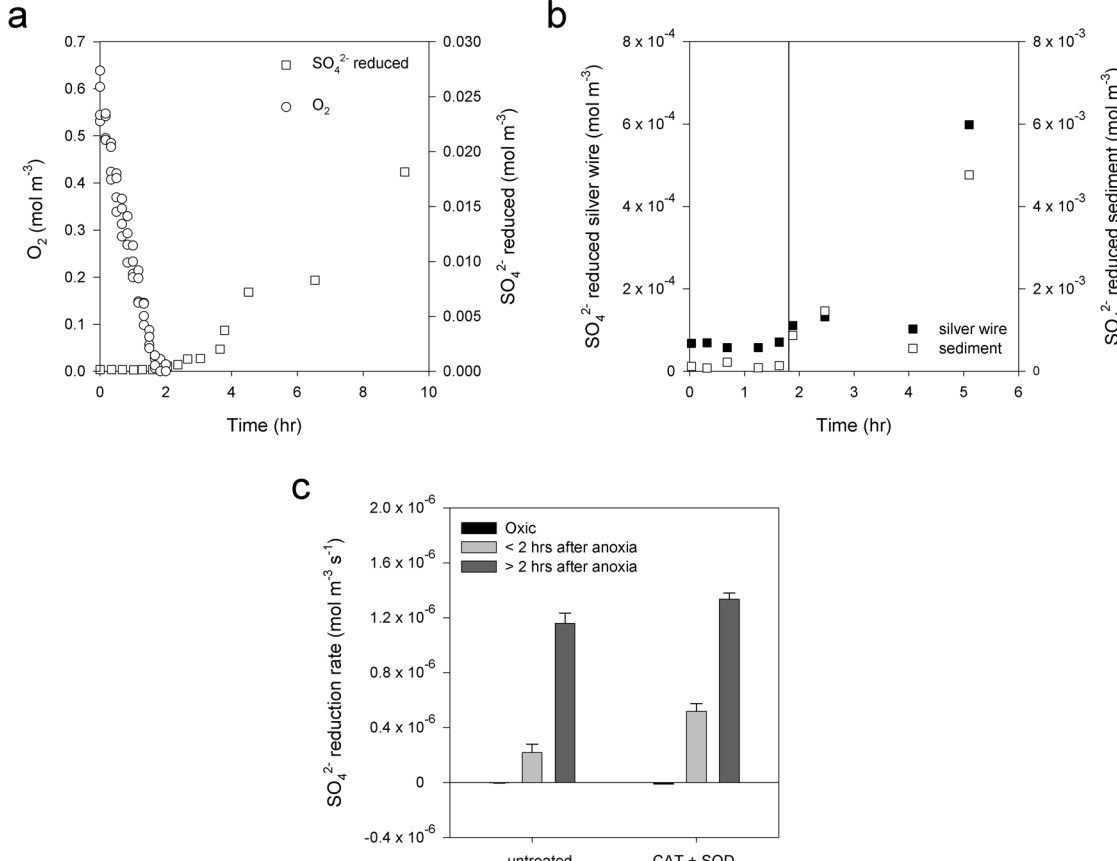

**Fig. 4 | Influence of oxygen (O₂) on respiration. a** Oxygen concentrations (open circles) and sulfate (SO₄²⁻) reduced (open squares) in mol m⁻³ sed in slurries over an oxic-anoxic transition. Slurries were from sediment collected on May 25, 2020. Oxygen concentrations are shown for four Exetainers, resampled over the course of the incubation. **b** Radiolabeled sulfate reduced bound to a silver wire (closed squares), or to the sediment in the same incubation vial (open squares) in incubations including a silver wire over an oxic-anoxic transition. For all sulfate reduction measurements, each point corresponds to a separate incubation vial. The vertical line represents the transition to anoxic conditions. **c** Sulfate reduction rates in sediment slurries from May 25, 2020 during three different periods of incubation. Slurries were untreated or amended with a combination of catalase (CAT) and superoxide dismutase (SOD). Black bars represent the sulfate reduction rate in the oxic period of the incubation, light gray bars represent the sulfate reduction rate between the start and 2 h after the start of anoxia, dark gray bars the sulfate reduction rate between 2 h after the start of anoxia and the end of the incubation (20–22.5 h). Error bars represent the standard error of the slope.

sulfide reoxidation to intermediate sulfur oxidation states, e.g., via reactions with iron oxides, would also have been detected by the radiochemical reduction method used[50]. Sulfate reduction rates in slurries treated with a combination of catalase and superoxide dismutase were higher mainly directly upon anoxia (Fig. 4c), suggesting that recovery of sulfate reducers was faster when ROS were removed. Thus, while sulfate reduction was controlled by oxygen, the sulfate reducers were not killed, but instead inactive. The rapid recovery upon anoxia suggests that microbes are robust to ROS, while our incubations suggest that ROS reduce respiration. Microbes may have substantial ROS defenses that allow rapid respiration after anoxia, but are not sufficient to overcome all ROS inhibition. Similarly, ROS has been shown to be responsible for bacterial inactivation and changes in microbial community structures in Fe(II)-containing sediments from a more stable river-groundwater system[51].

Changes in hydrogen levels were minor in the first 24 h of incubation, with levels remaining below 1 nmol cm⁻³ sed (Supplementary Fig. 10). Neither Fe²⁺ (Supplementary Fig. 11) nor methane (Supplementary Table 5) accumulated immediately after anoxia. Fe²⁺ accumulation started after more than 10 h, while methane concentrations stayed constant below 2 nmol cm⁻³ sed during the first 24 h. While this lag period could indicate that fermenters, iron reducers, and methanogens are more sensitive to oxygenation than sulfate reducers, these communities are regularly exposed to oxygen, so under selection pressure, are probably adapted to oxygen exposure. Rather than

anaerobic processes being stalled for so long, hydrogen, reduced iron, and methane are likely rapidly re-oxidized by pools of electron acceptors, such as Fe(III) and manganese(III,IV)[52], until these pools are exhausted[53]. Sulfide was also scavenged by these pools of oxidants as concentrations were very low (Supplementary Table 6), despite the occurrence of sulfate reduction.

## Impact of ROS on mineralization in intertidal sediments
Here, we show that removal of extracellular ROS within intertidal permeable sediments substantially boosts the potential rates of oxygen consumption, sulfate reduction, and Fe²⁺ and hydrogen accumulation. Aerobic respiration and sulfate reduction together are responsible for most of the mineralization in coastal sediments[32], so factors limiting these processes have the potential to directly impact the effectiveness of sands as biocatalytic filters (Fig. 5). We propose that ROS reduce biotic mineralization either through abiotically changing the availability of organic carbon, such as the direct degradation of organic molecules by ROS[13], or due to biotic effects such as through direct oxidative stress[6]. While this study points to the importance of ROS in marine sediment biogeochemistry, the extent of the impact of ROS requires in situ measurements of these compounds over space and time. Yet, there is only very limited data on the potential for ROS formation in marine sediments and their in situ distribution[3,12], limiting our understanding of the importance of ROS in sediment biogeochemistry.

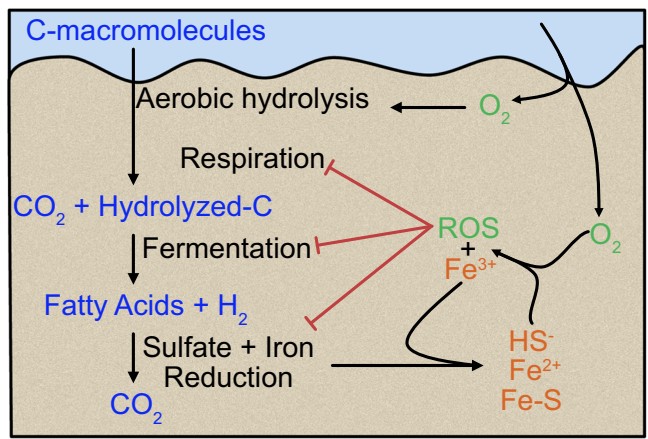

**Fig. 5 | Flow diagram of the proposed effects of reactive oxygen species (ROS) in intertidal sands.** Arrows represent transport and production processes. Red lines represent limiting effects. Organic material in the form of macromolecules is transported into the sediments. Hydrolysis and fermentation convert these molecules into e.g., fatty acids and hydrogen, which are substrates for sulfate and iron reducers. Also, oxygen ($O_2$) is transported into the sediments, where it can come in contact with reduced sulfur and iron, resulting in, among others, ferric iron ($Fe^{3+}$) and ROS. ROS influence biotic reactions with organic material. Sulfate- and iron reduction are limited either indirectly via fermentation, or directly, which is not further assessed in this study.

## Methods

### Sampling

Sediments were collected in the German Wadden Sea from the back-barrier area behind the island of Spiekeroog, on the intertidal sandflat Janssand (53°44'25.51"N, 7°41'28.63"E)[27–29]. The flat is subjected to advective flushing according to the tides, which are semi-diurnal and have a tidal range of ~2 m. The flat is inundated at high tide and exposed for ~6 h around low tide. Samples were collected from the upper, sandy part of the flat during low tide on five occasions over three seasons, May 25th, June 15th, July 28th, October 8th, 2020, and March 18th, 2021. The upper part of the flat has a porosity of 0.33, a mean grain size of 176 μm, and a permeability of ~7.2 × 10⁻¹² m² (see refs. [27,28]). Sediment was scraped from the upper 2 cm, transferred into a canister and covered with seawater. Deeper sediments and depth profiles were collected with core liners. Seawater was collected adjacent to the flat. On March 18, 2021, porewater samples were sampled in situ for $Fe^{2+}$ and hydrogen peroxide measurements using Rhizons (Rhizosphere Research Products, The Netherlands). Sediment cores were sampled using core liners for oxygen and hydrogen peroxide measurements.

### Incubation set up

All incubations were carried out in 6-mL gas-tight vials, hereafter called Exetainers (Labco, UK) that were filled without headspace, with 2 cm³ sediment and 4 mL seawater. During incubation, all Exetainers were placed in light-impermeable roller tanks and inverted every 30 s, to allow thorough slurry mixing. All incubations, except for incubations in October 2020, were started on the same day as the sediment was sampled. In October 2020, sediment and seawater were stored at 4 °C for six days before incubations started. Incubations were conducted at room temperature. Exetainers were filled on a lab bench after sediment was thoroughly mixed, and seawater was shaken to bring to equilibrium with the air. For all measurements except oxygen, Exetainers were destructively sampled, so each timepoint represents a separate individual Exetainer.

Depending on the experiment, incubations were untreated, or amended with 28 mmol L⁻¹ sodium molybdate, 1500 U mL⁻¹ catalase, 217 U mL⁻¹ superoxide dismutase, a combination of 217 U mL⁻¹

superoxide dismutase and 1500 U mL⁻¹ catalase, or 0.5 mg mL⁻¹ bovine serum albumin (BSA). The BSA treatment served as a control for the enzyme treatment, to allow the separation of the effects of enzymatic activity from the effect of added protein. Separate Exetainers were used to measure sulfate reduction, hydrogen accumulation, and methane accumulation, depending on the experiment. When total sulfate and dissolved iron and sulfide were measured, these were measured from the supernatant of the hydrogen Exetainers.

### Oxygen consumption measurements

Oxygen concentrations were measured repeatedly from a series of 3–4 Exetainers which were quickly opened at regular intervals to allow insertion of a Clark-type oxygen microsensor produced in-house[54]. When this process introduced a bubble to the Exetainer headspace, the Exetainer was discarded. Oxygen microsensors were calibrated against air-saturated seawater and anoxic sodium ascorbate.

One linear trend line per treatment was plotted through the individual measurements of the Exetainers, and its intersection with an oxygen concentration of zero μM was calculated, which was defined as the transition between oxic and anoxic conditions. For each trend line, the standard error of the slope was calculated.

### Sulfate reduction rate measurements

Sulfate reduction was determined according to ref. [50]. In all, 250 kBq of ³⁵S-SO₄²⁻ were added to each Exetainer used for sulfate reduction measurements. Incubations were stopped by transferring the entire content of the Exetainers to 6 mL 20% (w/v) ZnAc and then stored at −20 °C until distillation. Reduced sulfur was distilled from samples using a cold acid-chromium distillation within 2 months. All bioactive extracellular sulfur, except for sulfate, should be captured in this fraction. Radioactivity in the distilled sulfur fraction was determined with a scintillation counter (Perkin-Elmer Tri-Carb 4910 TR; using Ultima-Gold Scintillation cocktail). Sulfate reduction rates were calculated by plotting a linear trend line through the individual measurements of the anoxic period, and for each trend line, the standard error of the slope was calculated. For incubations conducted in May 2020, sulfate reduction in untreated slurries and slurries treated with a combination of catalase and superoxide dismutase were calculated for different periods of the incubation (oxic, <2 h after anoxia, and >2 h after anoxia).

In July 2020, sulfate reduction was also determined as above with the inclusion of a silver wire twice the length of the Exetainer, to increase sensitivity to oxic sulfate reduction[49]. Exetainers were sampled as above, with a higher resolution during the oxic period. The silver wire was then rinsed twice in 50 mM sodium sulfate, then radioactivity was determined using a scintillation counter.

### Hydrogen measurements

At each timepoint during the incubation, a 2-mL headspace was created in an Exetainer using nitrogen gas by removing 2-mL supernatant. The Exetainer was shaken vigorously for 2 min to allow for headspace equilibration, then 1 mL of the headspace was injected into a gas chromatograph (Peak Performer RCP 910-Series, Peak Laboratories, USA) using a gas- and pressure-tight syringe. The gas chromatograph was calibrated against a 100 ppm hydrogen standard (Air Products, Germany). Hydrogen accumulation rates were calculated by plotting a linear trend line through the individual points, and for each trend line, the standard error of the slope was calculated.

### Dissolved iron, sulfide, and sulfate measurements

The supernatant that was replaced by $N_2$ during headspacing from the Exetainers for hydrogen determination was used to measure dissolved iron, sulfide, and sulfate. Immediately after removing the supernatant using a syringe, the syringe was connected to a 0.2-μm PTFE filter. The first 0.5 mL from the syringe was transferred directly into 0.1 mL 5% (w/

v) ZnAc for subsequent sulfide and/or sulfate analysis. In all, 1 mL from the remaining volume was added directly to 0.1 mL ferrozine for subsequent dissolved iron measurements. Dissolved iron was measured spectrophotometrically. Porewater samples collected using Rhizons were fixed with ferrozine (for $Fe^{2+}$; March 2021) and ZnAc (for sulfide; May 2020) and transferred to cuvettes and measured using spectrophotometric methods. Sulfide was measured spectrophotometrically using the methylene blue method[55], and dissolved iron using the ferrozine method[56]. Sulfate was measured using an ion chromatograph (Metrohm 920 Compact IC Flex, Metrohm AG, Switzerland) with a zinc trap, calibrated against a standard curve of a sulfate standard.

## Methane measurements
Slurries in Exetainers in July 2020 used for methane analysis were fixed using 200 μL saturated $ZnCl_2$ solution, and stored upside-down until analysis. A headspace of 2 mL was created using helium gas, and 500 μL headspace was injected in a gas chromatograph using a gas- and pressure-tight syringe. The gas chromatograph was calibrated against a 100 ppm methane standard (Air Liquide, Germany).

## Solid-phase iron extraction
Solid-phase iron was extracted from sediments in May and July 2020. Samples of ~100–500 mg taken from the sediment surface (0–2 cm depth), or from a depth of 10–14 cm in a freshly sliced core, were quickly transferred to 0.5 M HCl and allowed to react for 0.5 h. The extract was then immediately filtered through 0.2 μm PTFE syringe filters and analyzed spectrophotometrically using the ferrozine method[56].

## Hydrogen peroxide sensor
The sensor consisted of an etched 50 μm-thick platinum anode plated with platinum chloride (8% $PtCl_4$ in MilliQ water), an etched 100 μm-thick platinum guard, and a thick platinum reference. The anode, guard, and reference were mounted in a glass casing, with the sensing anode at a distance of ca 50 μm from the tip. The tip diameter of the outer capillary had a diameter of 25–30 μm and a tip opening of 10 μm. Before mounting the electrodes, the tip of the outer capillary was sealed by a thin polyurethane membrane (D6)[57]. The membrane was dissolved in tetrahydrofuran (50 mg mL$^{-1}$) and applied by shortly immersing the capillary in the solution that is kept in the tip of a Pasteur pipette and left to cure overnight. The membrane was applied under microscopic guidance. The membrane separated the electrolyte from the seawater but was permeable for hydrogen peroxide. After mounting the electrodes in the casing, the sensor was filled with electrolyte, a borate/potassium chloride buffer (50 mM borate, 3 M potassium chloride and 500 μM ferrozine), with pH 9. Sensor performance is described in the Supplementary Information.

The sensor was connected to a picoammeter and polarized at +700 mV until a stable current was obtained, which happened normally within an hour. The medium in which the sensor was used was connected to an external reference electrode. The sensors were calibrated before use in a stirred beaker with filtered seawater to which aliquots of stabilized 3% hydrogen peroxide were added.

## Hydrogen peroxide microsensor measurements
Steady-state hydrogen peroxide microprofiles were measured using the new hydrogen peroxide microsensor (see Hydrogen peroxide sensor, Supplementary Information, Supplementary Table 7, and Supplementary Figs. 12 and 13) in a sediment core collected March 18, 2021. Parallel oxygen microprofiles were measured using an oxygen microsensor as described in ref. [54]. The interface between the overlying water column and sediment was set at depth zero. The water column was continuously stirred to ensure a well-mixed column and a constant boundary layer. The oxygen microsensor was 2-point calibrated (air

and 1 M Na-ascorbate pH 11). The hydrogen peroxide microsensor was calibrated by incremental addition of a 3% hydrogen peroxide solution to seawater. Microprofiles were measured using a motor-equipped micromanipulator, controlled by a laptop on which also the data were acquired.

Production of hydrogen peroxide was calculated using the steady-state depth profile. Fluxes were calculated by multiplying the effective molecular diffusion coefficient ($D_{eff} = 1.18 \times 10^{-8}$ m$^2$ s$^{-1}$) with the concentration gradient, where $D_{eff} = D_0(1.13 \times 10^{-9}$ m$^2$ s$^{-1}) \times$ porosity$^{-2}$. Conversion (production or consumption) was then calculated using the change in flux over depth.

The dynamics of hydrogen peroxide and oxygen were assessed by the addition of oxygen-saturated seawater and hydrogen peroxide solution to sediment in parallel cores. A needle connected to a syringe filled with the seawater, or seawater containing hydrogen peroxide, was slowly inserted into the sediment. Care was taken that injection occurred close to the sensors. Concentrations of hydrogen peroxide and oxygen were measured over time, while keeping the sensors at constant depth (3 or 4 cm).

## Hydrogen peroxide chemiluminescent measurements
Hydrogen peroxide concentrations from sediment porewater were determined in a FeLume system[58,59], essentially a circular flowcell with a photomultiplier placed directly on top to detect the photons from a chemiluminescent reaction in the flowcell. The flowcell and detector combination were placed in a black box to protect against light interference. Hydrogen peroxide is proportional to the number of photons produced by the chemiluminescent reaction with 10-methyl-9-(p-formylphenyl)-acridinium carboxylate trifluoromethanesulfonate (AE) under alkaline conditions[58,59]. The analysis was carried out in a flow-injection mode. Reagent solutions were prepared in 18.2 MΩ-cm MilliQ water with analytical-grade reagents. A pH 3 phosphate buffer with freshly 2 μM AE reagent served as the sample carrier solution. A 6-way injector valve with 50 μL sample loop was used to inject the samples and standards in the carrier stream. Both carrier stream and a 0.1 M sodium carbonate solution (pH 11.3) were pumped using a peristaltic pump (Gilson Minipuls 3) at flow rates of 2 mL min$^{-1}$ into the flowcell. Directly upon mixing the carrier flow and the alkaline buffer the chemiluminescent reaction started.

Catalase (10 U mL$^{-1}$) was added to AE and carbonate reagents to remove background hydrogen peroxide. The reagents with added catalase were left for a couple of hours which helped to obtain a stable baseline for the assay. For calibration, standard solutions (0.5–50 μM) of hydrogen peroxide were prepared in aged 0.22 μm-filtered seawater collected from the sampling area. Following extraction, porewater samples were immediately injected into the running FeLume. The response time to obtain a chemiluminescent peak was -15 s after the injection. Standards were injected periodically during the assays, to check for drift. Hydrogen peroxide standards with added catalase (100 U mL$^{-1}$) were also injected to confirm the disappearance of the signal during the assay.

Porewaters contain high concentrations of $Fe^{2+}$ (Supplementary. Fig. 3). Oxidation of $Fe^{2+}$ during sampling and analysis can lead to hydrogen peroxide generation. To prevent such interference by $Fe^{2+}$, we added ~200 μM ferrozine to the sampling syringe while drawing 2–3 mL of porewater using Rhizons (Rhizosphere Research Products, The Netherlands). Ferrozine was also added to the reagents and standards at similar concentrations to prevent any bias. Ferrozine does not have any effect on the hydrogen peroxide concentrations[58].

## Hydrogen distributions
In July 2020 and March 2021, cores were taken from the upper flat using core liners with 1 cm ports drilled into the side every 2 cm. On the sandflat (or in the harbor in March 2021), 12 mL Exetainers were

filled with 10 mL 35% (w/v) NaCl solution (6 mL Exetainers with 4 mL 35% (w/v) NaCl solution in March 2021), in a modification of the procedure outlined in ref. [60]. 2 cm$^3$ of sediment were removed from the cores every 2 cm via the ports using cut-off syringes and transferred into the Exetainers. Exetainers were capped without headspace. Four Exetainers served as controls and were filled with 35% (w/v) NaCl solution to account for any hydrogen produced during transport. Immediately upon returning to the home laboratory in Bremen (~3 h later) 2 mL of the water phase of each Exetainer was replaced with nitrogen gas. After equilibration, headspace hydrogen concentrations were measured using a gas chromatograph (see hydrogen measurements). The hydrogen present in the control Exetainers was subtracted from the amounts in the Exetainers with sediment added, after correcting for the slightly increased water volume in the control Exetainers.

## Carbon turnover

Carbon turnover rates for untreated slurries and slurries treated with a combination of catalase and superoxide dismutase were used to assess the influence of ROS on carbon turnover in the sediments. Carbon turnover was calculated using the aerobic respiration and sulfate reduction rates for slurries from sediment collected in June 2020. Linear trend lines were plotted through the individual oxygen measurements and reduced sulfate measurements for untreated slurries, and slurries treated with a combination of catalase and superoxide dismutase. For sulfate reduction rates, only the anoxic period of the incubations was used. A stoichiometry of sulfate:carbon of 1:2, and of oxygen:carbon of 138:106 was used to convert these rates to carbon turnover rates. As sulfate reduction is the dominant anaerobic pathway in coastal marine sediments, summed carbon turnover rates derived from aerobic respiration and sulfate reduction are a reliable estimation of the total carbon turnover rate via biotic respiration.

## Data availability

Data that support the findings of this study are available within the paper and its supplementary files. Source data are provided with this paper and are available in PANGAEA at https://doi.org/10.1594/PANGAEA.955443.

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

## Acknowledgements

We thank Gaby Eickert, Karin Hohmann, Vera Hübner, Anja Niclas, Ines Schröder and Cäcilia Wigand for their role in the development of the hydrogen peroxide sensor, sensor construction, and technical assistance. Volker Meyer is thanked for building the FeLume system and creating its software, Gunter Wegener for assistance with hydrogen measurements, Elisa Merz for help during sampling, and BTS Bootstouren Spiekeroog for transport. We thank Tim Ferdelman for fruitful discussions. We thank Anders Tjell for providing the polyurethane membrane. This study was funded by the Max Planck Society and the National Science Foundation (CMH; NSF OCE-1924236).

## Author contributions

Conceptualization: M.v.E., O.B., and D.d.B.; investigation: M.v.E., O.B., C.M., S.B., and D.d.B.; methodology: M.v.E., O.B., C.H., and D.d.B.; visualization: M.v.E. and O.B.; writing—original draft: M.v.E. and O.B.; writing—review & editing: M.v.E., O.B., C.M., S.B., C.H., and D.d.B.

## Funding

## Competing interests

The authors declare no competing interests.
