## [Peer Review File · Nature Communications]

Reactive oxygen species affect the potential for mineralization processes in permeable intertidal flatsReviewer #2 (Remarks to the Author):

In the submitted manuscript, the authors investigate the presence of reactive oxygen species (ROS) in intertidal flat sediments. These ROS, generally comprising of superoxides, hydroxyl radical, and peroxides, may be formed in the presence of organic matter, iron, sulfur species and may damage cells, inhibiting microbial activity. Given the geochemistry of these intertidal flats, the hypothesis that ROS may be forming is solid.

The authors performed some clever experiments – developing a new H₂O₂ sensor, using enzymes that quench the effect of ROS – to investigate the effect of these species on microbial respiration. Clearly, the authors have put in a lot of hard work to use good methods for this study. The results from the experiments performed in the sections “Impact of reactive oxygen species on respiration” and “Hydrogen peroxide in intertidal sediments” are compelling and of use to the wider scientific community. The development of an H₂O₂ sensor will be particularly useful as this is not an easy task. However, there are few key issues in the manuscript as it is written, especially the section “Anaerobic respiration after transient oxygenation”, that bring the overall conclusions of the manuscript into some doubt. I list out my key concerns below, followed by some minor comments:

1) Lack of context in the writing:

As soon as the first result started, I was immediately wondering – are these slurry experiments? Column incubations? If they are anaerobic, where does the oxygen come from? The complete lack of context threw me off and made it difficult to figure out what conditions were being tested. I understand that the materials and methods are at the end in this kind of manuscript, but in the present manuscript there is no frame at all in which to read the results. Please add some context either to the end of the introduction or the start of the results. For example, it would be useful to have a couple of sentences to lead into the results saying that we first tested sediment from X under Y conditions by adding the two enzymes, then complemented these results with direct measurements of peroxide in the porewater. Given the Fe content (A-B%) and low levels of sulfide, we expected...

2) Effect of using catalase and superoxide dismutase:

The results shown in Figure 1a are clear. The addition of both enzymes (necessary to completely remove ROS) results in a big jump in O₂ consumption rates. The results shown in figures 1b, 1c, and 1d are less clear: if both enzymes were necessary to remove ROS, why is there very similar sulfate reduction, ferrous iron, and hydrogen production when only catalase was added? For example, in the hydrogen production figure, it seems as if adding the catalase (i.e., quenching peroxide) was enough to stimulate hydrogen production. In that case, do the authors think that there isn't any superoxide (or effect of superoxide)?

Further, if the catalase and superoxide dismutase produce oxygen, why do we still see Fe²⁺? Is the oxygen produced too little for oxidation of the Fe²⁺? Certainly, some more discussion is required here.

3) Section “Anaerobic respiration after transient oxygenation”:

By this point in the manuscript, I was quite convinced that there are ROS in the porewater of intertidal flats and they affect anaerobic respiration. The results of this section however suggest that under oxic-anoxic fluctuations, any ROS that are produced actually don't affect respiration. The authors even say that fermenters, iron reducers, and methanogens are likely not affected adversely by the presence of oxygen. This leaves the reader wondering: are ROS then actually important? If by one oxic-anoxic transition, there is no real effect on respiration, then over multiple fluctuations, it might be the same. However, the authors say in the implications section that the relevance of ROS will be amplified in fluctuating redox environments (lines 191-192). Based on my reading of the authors' results, I would interpret it as yes, ROS forms, but they are likely not playing a large role. Which, in itself, is a rather interesting result and worthy of communication to the scientific community.

Overall, I think this section requires some clarity and discussion. For example, the authors suggest that oxygen is required for hydrolysis of macromolecules in sediments (line 172) and ROS could be breaking down macromolecules to lower weight

(considered more bioavailable) molecules (lines 173-175). These are possible explanations which the authors should expand upon. Further, the authors should make sure that their final message is in keeping with their result that ROS may not actually inhibit respiration in the environment.

Is there a control experiment they could do with the ROS quenched during the oxic-anoxic incubation that would show even higher sulfate reduction rates in the anoxic period, indicating that the ROS did affect microbial metabolisms? If this has been done, it was rather easy to miss even after going through the methods, underscoring my first point about lack of context.

Overall, I recognize the value of this work, but as presented, I cannot recommend its publication because the concerns above directly relate to the conclusions of the manuscript. I recommend that the authors revise their manuscript extensively, if possible including the control experiment.

Minor comments:

Lines 78-79: is the fact that superoxide dismutase acts on superoxide in a proton consuming manner relevant? What is the effect of the pH (given that seawater is slightly alkaline) on this process?

Lines 100-109: nicely shown!

Lines 158-159: do you mean no Fe²⁺ is immediately observed? In the next sentence, you say that Fe²⁺ does accumulate after 10 hours

Lines 149-155: the writing here could be tighter - these three-four sentences switch back and forth between what is expected and what actually was observed. Some editing in this instance would be helpful because this is quite an important point.

Lines 194-196: I think the language is too strong here. This study does not actually show that there are extra electron donating compounds due to an oxic-anoxic transition; this is one possible reason for the lack of inhibition due to ROS. Please soften the language here. Also, I recommend against using the phrase "handsomely rewarded with..".

Reviewer #3 (Remarks to the Author):

The authors describe the effect of transient oxygenation on microbial degradation processes in slurries made of seawater and intertidal surface sediment. The authors hypothesize that these sediments have a large capacity to produce reactive oxygen species (ROS) because of shifting oxic-anoxic interfaces and intense iron-sulfur cycling. Addition of catalase and superoxide dismutase enzymes increased rates of aerobic respiration, sulfate reduction and hydrogen accumulation in the slurries, and the authors concluded that ROS have a stifling effect and may strongly influence biogeochemistry of intertidal sediments with transient oxygenation events.

The spectrum of analyses done in this study is impressive. The results from the comprehensive slurry experiments are novel and interesting, and if the effect of ROS on sedimentary microbial activity could be demonstrated in the natural environment, this process could have far-reaching consequences for the cycling of matter. A new microsensor was introduced to measure hydrogen peroxide profiles in sediment cores, however, all results are based on laboratory measurements. My concern with this study is whether the main conclusions are justified based on the laboratory experiments.

It is plausible that ROS may have a stifling effect on aerobic respiration and sulfate reduction, but it is questionable whether the magnitude of this effect can be realistically determined from the slurry experiments. The slurries, composed of 2 cm³ sediment and 4 mL seawater were inverted every 30 seconds, causing continuous mixing, and preventing the compaction of the sediment. Such settings have the potential to strongly enhance microbial reaction rates and may not reflect the rates in the intertidal sands.

The oxygen microprofile measured in a retrieved sediment core exposed to continuous stirring and a well-mixed column (Fig. 2) shows an oxygen penetration of a couple of millimeters, suggesting that the mixing and oxygenation of the sediment at the study site may be less effective than in the slurry vials.

Temperature is a key factor controlling microbial reactions, and temperature is not reported for the slurry experiments. Assuming that these experiments were conducted at room temperature, this could bias the reaction rates calculated from the slurry incubations. The authors should add temperature information and discuss the potential impact on the results.

A comparison between the rates determined in the slurries and rates determined in-situ would be necessary to relate the laboratory data to the field and to demonstrate the relative influence of ROS on the intertidal biogeochemistry.

Additional comments:

The authors show one hydrogen peroxide profiles measured in a sediment core to demonstrate that hydrogen peroxide can build up in the anoxic sediment layer. Since this buildup is central for estimating the potential influence of ROS on microbial activities, the authors should include more profiles and concentration ranges.

Reactive oxygen species may react with organic matter within the sediment which can increase or reduce their lifetime. For the interpretation of the data, it would be helpful if the authors could add information on the lifetime of the hydrogen peroxide and its temporal dynamics (e.g., from profiles).

Similar amounts of organic matter were added to the slurries in form of catalase, superoxide dismutase or bovine serum albumin (BSA). The resulting concentration of protein in the slurries may have reached the order of magnitude of organic matter naturally present in the intertidal sandflat sediment. One could expect that the microbial community responds to that organic matter addition with increased degradation rates, irrespective whether it is BSA or enzymes. The authors state that incubations with a bovine serum albumin did not stimulate the process rates of oxygen consumption, sulfate reduction, and Fe²⁺ and hydrogen accumulation. Since bovine serum albumin is highly degradable, the question arises why no increases in microbial degradation rates were recorded. In contrast, catalase and superoxide dismutase caused significant increases, and because no significant response was recorded for the serum albumin addition, the authors concluded that the removal of ROS facilitated the higher degradation. To support that conclusion, the range of degradation rates that could be caused by the microbial decomposition of the catalase and superoxide dismutase in slurries not containing ROS should be reported.

We thank both reviewers for their constructive comments on our manuscript. Our responses to the reviewer comments below are in green.

REVIEWER COMMENTS

Reviewer #2 (Remarks to the Author):

In the submitted manuscript, the authors investigate the presence of reactive oxygen species (ROS) in intertidal flat sediments. These ROS, generally comprising of superoxides, hydroxyl radical, and peroxides, may be formed in the presence of organic matter, iron, sulfur species and may damage cells, inhibiting microbial activity. Given the geochemistry of these intertidal flats, the hypothesis that ROS may be forming is solid.

The authors performed some clever experiments – developing a new H₂O₂ sensor, using enzymes that quench the effect of ROS – to investigate the effect of these species on microbial respiration. Clearly, the authors have put in a lot of hard work to use good methods for this study. The results from the experiments performed in the sections “Impact of reactive oxygen species on respiration” and “Hydrogen peroxide in intertidal sediments” are compelling and of use to the wider scientific community. The development of an H₂O₂ sensor will be particularly useful as this is not an easy task.

However, there are few key issues in the manuscript as it is written, especially the section “Anaerobic respiration after transient oxygenation”, that bring the overall conclusions of the manuscript into some doubt. I list out my key concerns below, followed by some minor comments:

1) Lack of context in the writing:

As soon as the first result started, I was immediately wondering – are these slurry experiments? Column incubations? If they are anaerobic, where does the oxygen come from? The complete lack of context threw me off and made it difficult to figure out what conditions were being tested. I understand that the materials and methods are

at the end in this kind of manuscript, but in the present manuscript there is no frame at all in which to read the results. Please add some context either to the end of the introduction or the start of the results. For example, it would be useful to have a couple of sentences to lead into the results saying that we first tested sediment from X under Y conditions by adding the two enzymes, then complemented these results with direct measurements of peroxide in the porewater. Given the Fe content (A-B%) and low levels of sulfide, we expected...

We thank the reviewer for pointing this out – we realize now how our structure was disorienting. We added a few sentences to the last paragraph of the introduction section, discussing the aims, methods, and results of the study (L81-95). We also changed the order of paragraphs in the introduction section, and added an extra paragraph (L58-66). Additionally, we moved the section reporting the concentrations of the standing stock of hydrogen peroxide and hydrogen peroxide dynamics (L99-152) to in front of the section reporting the impact of ROS removal on incubation rates (L154-241). We furthermore added a few lines in each subsection of the results and discussion, which gives background and should more gradually lead to the results of the respective section (L101-105, L156-165, L273-282).

2) Effect of using catalase and superoxide dismutase:

The results shown in Figure 1a are clear. The addition of both enzymes (necessary to completely remove ROS) results in a big jump in O₂ consumption rates. The results shown in figures 1b, 1c, and 1d are less clear: if both enzymes were necessary to remove ROS, why is there very similar sulfate reduction, ferrous iron, and hydrogen production when only catalase was added? For example, in the hydrogen production figure, it seems as if adding the catalase (i.e., quenching peroxide) was enough to stimulate hydrogen production. In that case, do the authors think that there isn't any superoxide (or effect of superoxide)?

As superoxide is an extremely reactive molecule, much more reactive than hydrogen peroxide, we did not aim for quantitative superoxide measurements in this study. However, considering hydrogen accumulation and Fe²⁺ accumulation, slurries treated with SOD did show higher rates than untreated slurries. This points to the presence of superoxide. We added a sentence in which we provide an additional hypothesis why the effect of addition of SOD is minimum (L199-201).

Further, if the catalase and superoxide dismutase produce oxygen, why do we still see Fe²⁺? Is the oxygen produced too little for oxidation of the Fe²⁺? Certainly, some more discussion is required here.

We added a few lines discussing this in the manuscript (L214-218). The Fe^{2+} accumulation rates shown in the figure reflect net accumulation rates. We only see the build-up of Fe^{2+} hours after anoxia, and Fe^{2+} accumulation rates were calculated for the anoxic period. As Fe^{2+} reacts extremely quickly with oxygen, its presence in our incubation is good evidence that our incubations were completely anoxic. Likely Fe^{2+} first gets recycled, and only after complete anoxia (and clearance of ROS) is allowed to accumulate. Any oxygen produced by the activity of catalase and superoxide dismutase is likely directly used for oxidation of reduced compounds produced during the anoxic conditions of the slurry, such as Fe^{2+} and sulfide. The production rate of oxygen by enzyme activity was thus likely lower than the production rate of Fe^{2+} .

3) Section “Anaerobic respiration after transient oxygenation”:

By this point in the manuscript, I was quite convinced that there are ROS in the porewater of intertidal flats and they affect anaerobic respiration. The results of this section however suggest that under oxic-anoxic fluctuations, any ROS that are produced actually don't affect respiration. The authors even say that fermenters, iron reducers, and methanogens are likely not affected adversely by the presence of oxygen. This leaves the reader wondering: are ROS then actually important? If by one oxic-anoxic transition, there is no real effect on respiration, then over multiple fluctuations, it might be the same. However, the authors say in the implications section that the relevance of ROS will be amplified in fluctuating redox environments (lines 191-192). Based on my reading of the authors' results, I would interpret it as yes, ROS forms, but they are likely not playing a large role. Which, in itself, is a rather interesting result and worthy of communication to the scientific community.

We thank the reviewer for pointing this out – indeed this was not clear. We did not mean to say that the ROS that are produced don't affect mineralization, rather the contrary. Instead, the message we wanted to give is that sulfate reducers are not adversely affected in their anoxic respiration by oxygen, as a previous period of oxygenation does not lead to an absence of sulfate reduction during the subsequent anoxic period. This could be due to abiotic effects of ROS, by increasing the decomposition of larger molecules that can then be used by sulfate reducers. However, we understood that this section might be confusing, and decided to take this part out of the manuscript, do further experiments on the effect of ROS on fermentation and hydrolysis for another manuscript. We therefore also removed Figure 3C of the originally submitted manuscript.

Instead, we described in more detail why we expected sulfate reduction to take place under oxic conditions (L273-282). We added new Figure 4C, in which we go into more detail into the recovery of sulfate reducers after anoxia under the influence of

ROS (L299-305).

Overall, I think this section requires some clarity and discussion. For example, the authors suggest that oxygen is required for hydrolysis of macromolecules in sediments (line 172) and ROS could be breaking down macromolecules to lower weight (considered more bioavailable) molecules (lines 173-175). These are possible explanations which the authors should expand upon. Further, the authors should make sure that their final message is in keeping with their result that ROS may not actually inhibit respiration in the environment.

We decided to remove this section, including Figure 3C, since this data is difficult to interpret with the current dataset and needs further study. We will expand on this topic in a subsequent study.

Is there a control experiment they could do with the ROS quenched during the oxic-anoxic incubation that would show even higher sulfate reduction rates in the anoxic period, indicating that the ROS did affect microbial metabolisms? If this has been done, it was rather easy to miss even after going through the methods, underscoring my first point about lack of context.

This experiment has been done, and is now added (Figure 4C), replacing Figure 3C in the original submission. The results of this experiment show that sulfate reduction rates are higher during the anoxic period when ROS are removed (especially during the first 2 hours after the transition to anoxia). Similarly, this is shown in Figure 2B, where the sulfate reduction rates were calculated over the anoxic period (as no sulfate reduction occurred during the oxic period). This is now more specifically mentioned (L194). We hope that with the additional background information provided at the beginning of each section, the context of the experiments is easier to follow.

Overall, I recognize the value of this work, but as presented, I cannot recommend its publication because the concerns above directly relate to the conclusions of the manuscript. I recommend that the authors revise their manuscript extensively, if possible including the control experiment.

Minor comments:

Lines 78-79: is the fact that superoxide dismutase acts on superoxide in a proton consuming manner relevant? What is the effect of the pH (given that seawater is slightly alkaline) on this process?

Seawater has an enormous buffering capacity, so it is unlikely that superoxide dismutase could have changed the pH of the seawater. Further, consumption of

protons would increase the pH, while production of CO₂ decreases the pH, and thus balances pH changes.

Lines 100-109: nicely shown!

Lines 158-159: do you mean no Fe²⁺ is immediately observed? In the next sentence, you say that Fe²⁺ does accumulate after 10 hours

Indeed, not immediately observed. Changed to: Neither Fe²⁺ nor methane accumulated immediately after anoxia (L314-316).

Lines 149-155: the writing here could be tighter - these three-four sentences switch back and forth between what is expected and what actually was observed. Some editing in this instance would be helpful because this is quite an important point.

We now introduced the observations with a paragraph explaining the background and aim of the experiment (L273-282), followed by a paragraph discussing our results (L284-311).

Lines 194-196: I think the language is too strong here. This study does not actually show that there are extra electron donating compounds due to an oxic-anoxic transition; this is one possible reason for the lack of inhibition due to ROS. Please soften the language here. Also, I recommend against using the phrase “handsomely rewarded with..”.

This section has been deleted, we decided to conduct further experiments on the effect of ROS on hydrolysis and fermentation.

Reviewer #3 (Remarks to the Author):

The authors describe the effect of transient oxygenation on microbial degradation processes in slurries made of seawater and intertidal surface sediment. The authors hypothesize that these sediments have a large capacity to produce reactive oxygen species (ROS) because of shifting oxic-anoxic interfaces and intense iron-sulfur cycling. Addition of catalase and superoxide dismutase enzymes increased rates of aerobic respiration, sulfate reduction and hydrogen accumulation in the slurries, and the authors concluded that ROS have a stifling effect and may strongly influence biogeochemistry of intertidal sediments with transient oxygenation events.

The spectrum of analyses done in this study is impressive. The results from the comprehensive slurry experiments are novel and interesting, and if the effect of ROS on sedimentary microbial activity could be demonstrated in the natural environment,

this process could have far-reaching consequences for the cycling of matter. A new microsensor was introduced to measure hydrogen peroxide profiles in sediment cores, however, all results are based on laboratory measurements. My concern with this study is whether the main conclusions are justified based on the laboratory experiments.

It is plausible that ROS may have a stifling effect on aerobic respiration and sulfate reduction, but it is questionable whether the magnitude of this effect can be realistically determined from the slurry experiments. The slurries, composed of 2 cm³ sediment and 4 mL seawater were inverted every 30 seconds, causing continuous mixing, and preventing the compaction of the sediment. Such settings have the potential to strongly enhance microbial reaction rates and may not reflect the rates in the intertidal sands.

We believe that the use of slurries can be justified as we were interested in the question if ROS could affect mineralization in intertidal sediments. For this purpose, it was important to have homogeneous conditions within the sediments, and to precisely determine the shift to anoxic conditions. This cannot be achieved in in situ measurements or sediment cores. We do recognize that rates we measured in our slurries can deviate from in situ rates. However, while slurry experiments have the potential to increase microbial metabolism, the sulfate reduction rates obtained in this study are below the maximum rates, while in the same order of magnitude, as obtained at this site in the same season (June) using the whole-core incubation technique. Our highest sulfate reduction rate was $7 \times 10^{-7} \text{ mol m}^{-3} \text{ s}^{-1}$, compared to $\sim 100 \text{ nmol cm}^{-3} \text{ d}^{-1}$ in June i.e. $1.15 \times 10^{-6} \text{ mol m}^{-3} \text{ s}^{-1}$ (Al Raei et al. 2009,). We added a paragraph explaining our choice for slurry experiments (L167-173).

References:

Al-Raei, A. M., Bosselmann, K., Böttcher, M. E., Hespeneide, B. & Tauber, F. (2009). Seasonal dynamics of microbial sulfate reduction in temperate intertidal surface sediments: controls by temperature and organic matter. *Ocean Dynamics* 59, 351-370. <https://doi.org/10.1007/s10236-009-0186-5>.

The oxygen microprofile measured in a retrieved sediment core exposed to continuous stirring and a well-mixed column (Fig. 2) shows an oxygen penetration of a couple of millimeters, suggesting that the mixing and oxygenation of the sediment at the study site may be less effective than in the slurry vials.

During high tide oxygen penetrates the sediment of the upper flat 1-2 cm (Jansen et al. 2009) and the sediment is subject to frequent mixing. This upper sediment is what we incubated in our experiments. Indeed, there may normally be microniches of hypoxic

sediment in situ, however we wanted to ensure even oxygenation in our incubations, so that we could be confident about whether or not processes occurred under both the oxic and anoxic conditions.

Based on the oxygen penetration depth presented by Jansen et al. (2009), the surface sediments (top 2 cm) that we used in our experiments do often experience oxygenation, while during low tide they may become anoxic. The mixing and oxygenation in the slurry vials might indeed be more efficient than in the field, however, both the oxic and anoxic conditions of our slurry incubations are not unusual for these sediments.

References:

Jansen, S., Walpersdorf, E., Werner, U., Billerbeck, M., Böttcher, M. E., De Beer, D. (2009). Functioning of intertidal flats inferred from temporal and spatial dynamics of O₂, H₂S and pH in their surface sediments. *Ocean Dynamics* 59, 317-332.
<https://doi.org/10.1007/s10236-009-0179-4>.

Temperature is a key factor controlling microbial reactions, and temperature is not reported for the slurry experiments. Assuming that these experiments were conducted at room temperature, this could bias the reaction rates calculated from the slurry incubations. The authors should add temperature information and discuss the potential impact on the results.

Incubations were indeed conducted at room temperature, this information is now added (L380). Mean temperatures for the months in which was sampled were ranging from around 4 to around 18 °C (Deutscher Wetterdienst (German Weatherservice); website accessed July 26, 2022).

May 2020:

<https://www.dwd.de/EN/ourservices/klimakartendeutschland/klimakartendeutschland.html;jsessionid=C8A85B3EE65E865093040FF739D90CB2.live21071?nn=519080>

June 2020:

<https://www.dwd.de/EN/ourservices/klimakartendeutschland/klimakartendeutschland.html;jsessionid=C8A85B3EE65E865093040FF739D90CB2.live21071?nn=519080>

July 2020:

<https://www.dwd.de/EN/ourservices/klimakartendeutschland/klimakartendeutschland.html;jsessionid=C8A85B3EE65E865093040FF739D90CB2.live21071?nn=519080>

October 2020:

<https://www.dwd.de/EN/ourservices/klimakartendeutschland/klimakartendeutschland.html;jsessionid=C8A85B3EE65E865093040FF739D90CB2.live21071?nn=519080>

March 2021:

<https://www.dwd.de/EN/ourservices/klimakartendeutschland/klimakartendeutschland.html;jsessionid=C8A85B3EE65E865093040FF739D90CB2.live21071?nn=519080>

Although this is different from room temperature in which the incubations were conducted, which could lead to slightly different rates (Al-Raei et al. 2009), we expect that this did not affect our results. We were interested in the potential rates in the incubations, and the differences between the different treatments.

As temperatures in this environment are variable over the year, we expect the organisms to be adapted to changes in temperature. Room temperature was not extreme compared to the temperature organisms can experience in situ, so we do not expect a cold or heat shock of the organisms. This is also visible in the rates that do not deviate much from the rates of oxygen consumption and sulfate reduction measured during these seasons (Werner et al. 2003; Al-Raei et al. 2009).

References:

Al-Raei, A. M., Bosselmann, K., Böttcher, M. E., Hespeneide, B. & Tauber, F. (2009). Seasonal dynamics of microbial sulfate reduction in temperate intertidal surface sediments: controls by temperature and organic matter. *Ocean Dynamics* 59, 351-370. <https://doi.org/10.1007/s10236-009-0186-5>.

Werner, U., Polerecky, L., Walpersdorf, E., Franke, U., Billerbeck, M., Böttcher, M., Ferdelman, T. & De Beer, D. (2003). Organic matter degradation processes in permeable sediments – methodological approaches. *Ber Forschungszentrum Terramare* 12, 122-125.

A comparison between the rates determined in the slurries and rates determined in-situ would be necessary to relate the laboratory data to the field and to demonstrate the relative influence of ROS on the intertidal biogeochemistry.

It was not our goal to determine the rates of the processes, but rather to study potential effects of ROS on these process rates. Nevertheless, rates of oxygen consumption and sulfate reduction that we determined in the slurries are of the same order of magnitude as determined by other studies in Janssand sediments.

We added a sentence in the text mentioning that rates in the manuscript are potential conversion rates, but that measured rates are within previously reported process rates (L171-173).

Additional comments:

The authors show one hydrogen peroxide profiles measured in a sediment core to demonstrate that hydrogen peroxide can build up in the anoxic sediment layer. Since this buildup is central for estimating the potential influence of ROS on microbial activities, the authors should include more profiles and concentration ranges.

We now included more microprofiles measured with the hydrogen peroxide microsensor (Fig. 1, Supplementary Fig 1). We furthermore added a paragraph on the dynamics of hydrogen peroxide (L136-152). Results in this paragraph are supported with new Figures (Fig. 2; Supplementary Fig. S5). We show the delicate balance between production and consumption of hydrogen peroxide in these sediments. Firstly, via injections of hydrogen peroxide into the sediments. Hydrogen peroxide is quickly consumed, and oxygen is produced. Furthermore, we show that injection of oxygenated seawater leads to transient peaks of hydrogen peroxide. Currently, we cannot explain the presence of hydrogen peroxide in the deeper, anoxic sediments. We hypothesize that Fe cycling is involved in the production of hydrogen peroxide in these anoxic zones, which we now mentioned in L138-143. We are currently working on this topic in a subsequent study, in which we also measured hydrogen peroxide concentrations in another intertidal sandflat using microsensors and chemiluminescence (where we took the instruments on the sandflat), and also found that concentrations here are in the μM range.

Reactive oxygen species may react with organic matter within the sediment which can increase or reduce their lifetime. For the interpretation of the data, it would be helpful if the authors could add information on the lifetime of the hydrogen peroxide and its temporal dynamics (e.g., from profiles).

We added a paragraph discussing the dynamics of hydrogen peroxide in the sediments (L136-152). Hydrogen peroxide dynamics were studied via the injection of oxygenated seawater and hydrogen peroxide-containing oxygenated seawater close to oxygen and hydrogen peroxide microsensors.

Similar amounts of organic matter were added to the slurries in form of catalase, superoxide dismutase or bovine serum albumin (BSA). The resulting concentration of protein in the slurries may have reached the order of magnitude of organic matter naturally present in the intertidal sandflat sediment. One could expect that the microbial community responds to that organic matter addition with increased degradation rates, irrespective whether it is BSA or enzymes. The authors state that incubations with a bovine serum albumin did not stimulate the process rates of oxygen consumption, sulfate reduction, and Fe^{2+} and hydrogen accumulation. Since bovine serum albumin is highly degradable, the question arises why no increases in microbial degradation rates were recorded. In contrast, catalase and superoxide dismutase caused significant increases, and because no significant response was recorded for the serum albumin addition, the authors concluded that the removal of ROS facilitated the higher degradation. To support that conclusion, the range of degradation rates that could be caused by the microbial decomposition of the catalase and superoxide

dismutase in slurries not containing ROS should be reported.

Indeed, the amount of enzyme and BSA we added was substantial. The DOC of these sediments is 0.1-1% of their weight (e.g. Behrendt et al. 2013). At a density of about 2g/cm³, and 4 cm³ per incubation, we could have added 6-60% of the organic carbon already present in the sediment. However, since the oxygen consumption rate was only substantially increased with the addition of both catalase and superoxide dismutase, and not either enzyme alone, we are confident that the increased rate of oxygen consumption is related to enzymatic activity rather than degradation of the enzyme. We do not know why BSA did not stimulate the microbial community, and can only infer that it is not highly degradable in these time scales or by this community.

We are unsure how to calculate what degradation rates could be caused by the degradation of catalase and superoxide dismutase. If we simply increase the measured rates by 60%, to account for the added carbon, then only the sulfate reduction rates can be partially explained by the increased biomass from the enzymes. Further, the difference in sulfate reduction rates between catalase alone and the combination of catalase and superoxide dismutase suggests that enzyme activity is affecting the rates.

Alternatively, if we assume that the enzymes were 100% carbon (a substantial overestimate), i.e. 2.3 mg of carbon per incubation, or 190 μmol and 100% degraded within 24 hours (anoxic incubations) by 2 cm³ sediment, the carbon consumption rate would be far beyond what we measured – $1.1 \times 10^{-3} \text{ mol m}^{-3} \text{ s}^{-1}$.

References:

Behrendt, A., De Beer, D., and Stief, P. (2013). Vertical activity distribution of dissimilatory nitrate reduction in coastal marine sediments. *Biogeosciences* 10, 7509-7523. <https://doi.org/10.5194/bg-10-7509-2013>

Reviewer #2 (Remarks to the Author):

I appreciate that the authors took my comments into consideration and revised the manuscript. The study (both scientifically and with respect to writing) has greatly improved. Especially I appreciate that there is much more context in the writing – I enjoyed reading the revised manuscript. I thank the authors for providing clarifications in many places throughout the manuscript.

I have two major concerns with the current version, both of which have to do with how things are explained:

1. The interference of Fe²⁺:

The authors state on page 5, lines 88-90 that the microsensors contained ferrozine and were therefore not sensitive to Fe²⁺. This tripped me up – Fe²⁺ forms a complex with ferrozine (and the authors use this method to determine the Fe²⁺ concentration in the porewater). How is using ferrozine as an electrolyte not affecting the measurement in the presence of Fe²⁺? Looking at the Supplementary Information, specifically Figure 11b, it looks like the sensor is sensitive to Fe²⁺, with a slope that's >10% that of the H₂O₂ response. Given the high Fe²⁺ concentrations in the porewater compared to H₂O₂, it would follow that part of the signal may be due to Fe²⁺. This could also explain the 10 fold difference between the H₂O₂ concentration measured by microsensor and chemiluminescence.

It's also entirely possible that there is an explanation to the possible interference due to the Fe²⁺-ferrozine complex; if so, I suggest that the authors clearly put this in.

2. The effect of using catalase and superoxide dismutase:

The authors did address a couple of my previous comments in this section. The text in the lines 162-166 now specifies possible reasons for the apparent lack of effect of superoxide dismutase. However, the language here should likely still be softened. In the rebuttal, the authors say that slurries treated with superoxide dismutase showed higher accumulations than untreated slurries. However, looking at the error bars in Figure 3c and 3d, it's hard to say whether this is significantly higher. Especially since the line 162 specifies that superoxide dismutase alone did not have an effect. Based on the data shown, I would rather interpret that peroxide likely had a major role, but it is possible that superoxide was also present given the inhibition of superoxide dismutase by H₂O₂. Also, the rebuttal letter has the wrong line numbers – the lines 199-201 in the revised manuscript is about hydrogen accumulation.

Based on these two issues, I recommend acceptance after revision, specifically after the ferrozine-Fe²⁺ interference is explained.

A few minor comments:

Line 32: Is it normal that the first citation is numbered 7?

Line 65: Hypotheses are not confirmed, they are either supported or rejected.

Line 68: It's unclear what 'standing stock' means here. Does it mean that there are significant concentrations of H₂O₂ and H₂O₂ seems to be stable under these conditions?

Lines 107-123: Cool results!

Lines 191-201: Interesting results

Reviewer #3 (Remarks to the Author):

The revision improved the manuscript. The main conclusions of this study now are that ROS are formed in intertidal sediments where they modulate microbial mineralization rates. The direct measurements of the hydrogen peroxide profiles are novel. In the slurry experiments, brief sediment exposure to oxygen triggered higher subsequent sulfate reduction rates demonstrating strong effects of transient oxygenation and ROS on the incubated sediments biogeochemistry.

The rewrite of the section "Anaerobic respiration after transient oxygenation", now including a more detailed description of processes affecting sulfate reduction under oxic

conditions, is helpful. In this discussion section, I missed the reference to Ma et al. 2019 (ACS Earth Space Chem. 2019, 3, 738–747), who, in experiments with hydrogen peroxide and addition of ROS quencher, showed bacteria inactivation correlating with ROS produced upon Fe(II) oxygenation.

As stated in the response to the reviewers, the authors were interested in potential rates and the differences between treatments. Potential rates may differ substantially from actual in-situ rates. This should be reflected in the title of the manuscript, and the author should point this out to the reader. The rapid transition from anaerobic to aerobic settings in intertidal sediment during flooding requires pore fluid exchange which is not reproduced in the slurry incubations. Similar sulfate reduction rates measured in the field and in the slurries are not necessarily supporting that the slurries adequately mimicked field conditions. Slurries as used here are closed systems while the intertidal sediment is an open system that drains throughout ebb tide.

REVIEWER COMMENTS

We thank the reviewers for their positive and constructive comments on our manuscript. Our answers are written below each comment, in green.

Reviewer #2 (Remarks to the Author):

I appreciate that the authors took my comments into consideration and revised the manuscript. The study (both scientifically and with respect to writing) has greatly improved. Especially I appreciate that there is much more context in the writing – I enjoyed reading the revised manuscript. I thank the authors for providing clarifications in many places throughout the manuscript.

I have two major concerns with the current version, both of which have to do with how things are explained:

1. The interference of Fe²⁺:

The authors state on page 5, lines 88-90 that the microsensors contained ferrozine and were therefore not sensitive to Fe²⁺. This tripped me up – Fe²⁺ forms a complex with ferrozine (and the authors use this method to determine the Fe²⁺ concentration in the porewater). How is using ferrozine as an electrolyte not affecting the measurement in the presence of Fe²⁺? Looking at the Supplementary Information, specifically Figure 11b, it looks like the sensor is sensitive to Fe²⁺, with a slope that's >10% that of the H₂O₂ response. Given the high Fe²⁺ concentrations in the porewater compared to H₂O₂, it would follow that part of the signal may be due to Fe²⁺. This could also explain the 10 fold difference between the H₂O₂ concentration measured by microsensor and chemiluminescence.

It's also entirely possible that there is an explanation to the possible interference due to the Fe²⁺-ferrozine complex; if so, I suggest that the authors clearly put this in.

This is a good point, and crucial. We were not clear enough. The interference with Fe²⁺ occurs with a sensor that has an electrolyte without ferrozine (Supp. Fig. 12b). With 50 μM ferrozine in the electrolyte the interference stops, as tested with 20, 100 and 200 μM Fe²⁺ in N₂ flushed seawater brought to pH 3 (Supp. Fig. 2). We included Supp. Fig. 2 to clarify this. The range of Fe²⁺ concentrations tested cover the Fe²⁺ concentrations measured (Supp. Fig. 3). The low pH was chosen to assure the presence of Fe²⁺, as at this low pH undesired oxidation by O₂ is inhibited. We now routinely elevated the ferrozine content to 500 μM. Ferrozine does not change the H₂O₂ sensitivity. The methods are added in the Supplementary Methods section (Supplementary Information L171-174).

The difference with the extracts measured on the FeLume may be explained by losses during sampling and injections.

2. The effect of using catalase and superoxide dismutase:

The authors did address a couple of my previous comments in this section. The text in the lines 162-166 now specifies possible reasons for the apparent lack of effect of superoxide dismutase. However, the language here should likely still be softened. In the rebuttal, the authors say that slurries treated with superoxide dismutase showed higher accumulations than untreated slurries. However, looking at the error bars in Figure 3c and 3d, it's hard to say whether this is significantly higher. Especially since the line 162 specifies that superoxide dismutase alone did not have an effect. Based on the data shown, I would rather interpret that peroxide likely had a major role, but it is possible that superoxide was also present given the inhibition of superoxide dismutase by H₂O₂.

We softened the language by replacing the sentence “Superoxide dismutase alone had no effect” with “We found no evidence that superoxide dismutase could alone affect respiration processes” (L168-169). We furthermore added the sentence “Nevertheless, our data indicates that the addition of catalase, rather than superoxide dismutase, seems to be the driver for the increase of process rates after ROS removal” (L174-175).

Also, the rebuttal letter has the wrong line numbers – the lines 199-201 in the revised manuscript is about hydrogen accumulation.

We are sorry to have referred to incorrect line numbers.

Based on these two issues, I recommend acceptance after revision, specifically after the ferrozine-Fe²⁺ interference is explained.

A few minor comments:

Line 32: Is it normal that the first citation is numbered 7?

We are sorry for the numbering of the citations, this is now corrected.

Line 65: Hypotheses are not confirmed, they are either supported or rejected.

We changed “In this work, we confirm the hypothesis” to “This work supports our hypothesis” (L65).

Line 68: It's unclear what ‘standing stock’ means here. Does it mean that there are significant concentrations of H₂O₂ and H₂O₂ seems to be stable under these conditions?

Indeed, we did mean that concentrations of hydrogen peroxide are significant. We changed “a substantial stock of hydrogen peroxide” to “significant concentrations of hydrogen peroxide” (L68).

Lines 107-123: Cool results!

Lines 191-201: Interesting results

Reviewer #3 (Remarks to the Author):

The revision improved the manuscript. The main conclusions of this study now are that ROS are formed in intertidal sediments where they modulate microbial mineralization rates. The direct measurements of the hydrogen peroxide profiles are novel. In the slurry experiments, brief sediment exposure to oxygen triggered higher subsequent sulfate reduction rates demonstrating strong effects of transient oxygenation and ROS on the incubated sediments biogeochemistry.

The rewrite of the section “Anaerobic respiration after transient oxygenation”, now including a more detailed description of processes affecting sulfate reduction under oxic conditions, is helpful. In this discussion section, I missed the reference to Ma et al. 2019 (ACS Earth Space Chem. 2019, 3, 738–747), who, in experiments with hydrogen peroxide and addition of ROS quencher, showed bacteria inactivation correlating with ROS produced upon Fe(II) oxygenation.

Indeed, this is a relevant study. We now referenced to the study of Ma et al. 2019 in L246-248.

As stated in the response to the reviewers, the authors were interested in potential rates and the differences between treatments. Potential rates may differ substantially from actual in-situ rates. This should be reflected in the title of the manuscript, and the author should point this out to the reader. The rapid transition from anaerobic to aerobic settings in intertidal sediment during flooding requires pore fluid exchange which is not reproduced in the slurry incubations. Similar sulfate reduction rates measured in the field and in the slurries are not necessarily supporting that the slurries adequately mimicked field conditions. Slurries as used here are closed systems while the intertidal sediment is an open system that drains throughout ebb tide.

We changed the title of the manuscript to: Reactive oxygen species affect the potential for mineralization processes in permeable intertidal flats (L1-2).

We changed the abstract as follows:

“We furthermore investigate the effect of ROS on potential rates of microbial degradation processes in intertidal surface sediments after transient oxygenation, using slurries that transitioned from oxic to anoxic conditions. Enzymatic removal of ROS strongly increases potential rates of aerobic respiration, sulfate reduction and hydrogen accumulation.” (L20-24)

We now also mention specifically that we are studying potential rates in the last paragraph of the introduction:

“Potential biogeochemical process rates in slurries of these sediments amended with ROS-removing enzymes confirm that ROS can affect microbial respiration. Potential rates of oxygen consumption, sulfate reduction, and H₂ and dissolved Fe²⁺ accumulation are all increased by the removal of ROS.” (L70-73), and in the conclusion: “Here, we show that removal of extracellular ROS within intertidal permeable sediments substantially boosts the potential rates of oxygen consumption, sulfate reduction, and Fe²⁺ and hydrogen accumulation.” (L265-267).